META-RESEARCH

# The need for more research into reproductive health and disease

**Abstract:** Reproductive diseases have a significant impact on human health, especially on women's health: endometriosis affects 10% of all reproductive-aged women but is often undiagnosed for many years, and preeclampsia claims over 70,000 maternal and 500,000 neonatal lives every year. Infertility rates are also rising. However, relatively few new treatments or diagnostics for reproductive diseases have emerged in recent decades. Here, based on analyses of PubMed, we report that the number of research articles published on non-reproductive organs is 4.5 times higher than the number published on reproductive organs. Moreover, for the two most-researched reproductive organs (breast and prostate), the focus is on non-reproductive diseases such as cancer. Further, analyses of grant databases maintained by the Canadian Institutes of Health Research and the National Institutes of Health in the United States show that the number of grants for research on non-reproductive organs is 6–7 times higher than the number for reproductive organs. Our results suggest that there are too few researchers working in the field of reproductive health and disease, and that funders, educators and the research community must take action to combat this longstanding disregard for reproductive science.

**NATALIE D MERCURI AND BRIAN J COX***

## Introduction

It is difficult to overstate the impact of reproductive disease. Adverse pregnancy outcomes – which include preterm delivery, low birth weight, hypertensive disorders, and gestational diabetes –impact the acute and chronic health of the population (*Barker, 1997*; *Williams, 2011*; *Lewis et al., 2012*). About 20% of all pregnancies require medical intervention (*Murray and Lopez, 1998*), and in lower resource settings, pregnancy and delivery complications are a leading cause of maternal and neonatal death (*WHO, 2019*).

In 1992, the Institute of Medicine in the United States published a report called *Strengthening Research in Academic OB-GYN Departments* that outlined areas of research with obstetrics and gynecology where improvements were needed, such as low-birth-weight infants, fertility complications, and pregnancy-induced hypertension (*Institute of Medicine, 1992*). Three decades later, despite the essential nature and impact of the reproductive system, these issues are still major challenges in reproductive health.

Gender inequality and bias have been issues since the onset of biological and medical research. For example, including women as subjects in clinical research was not standard practice until after 1986 (*Liu and Mager, 2016*). There has been progress in developing policies to increase the representation of women (as both subjects and researchers) and in providing education on gender inequality for all researchers, but women are still underrepresented in scientific and medical research (*Huang et al., 2020*).

There are a variety of stigmas and taboos surrounding any topic relating to reproductive function. Menstruation is one function that has faced stigmatization that persists today (*Litman, 2018*; *Pickering, 2019*), with women often feeling too embarrassed to talk about this natural process or even complete an essential task, such as purchasing menstrual products at a local store. Political power highly affects reproductive health care and rights over other biological processes. In many countries, ongoing political and legal battles directly affect access to safe reproductive health care, including contraception, safe

***For correspondence:**
b.cox@utoronto.ca

**Competing interest:** The authors declare that no competing interests exist.

abortion, and gender identity rights (*Pugh, 2019*). There are parallels between the low level of research into reproductive diseases and the response to the AIDS epidemic in the 1980s. The long delay in recognizing AIDS as a significant health issue, and then implementing research policies, perpetuated false ideas surrounding the lifestyles of those affected by the disease and created a barrier to expanding sexual education and seeking healthcare, likely costing many lives (*Francis, 2012*). Despite great advances in AIDS research and treatment, including social awareness, public health stigma still lingers in society (*Turan et al., 2017*). Similar increases in advocacy and public awareness are needed to overcome these barriers affecting reproductive health.

Reproductive pathologies are often challenging to diagnose and properly treat, which increases the risk of comorbidity development. Moreover, a long-standing lack of research into reproductive health and disease means that the acute and chronic healthcare burden caused by reproductive pathologies is likely to continue increasing. This lack of research likely results from historic and ongoing systemic biases against female-focused research, and from political and legal challenges to female reproductive health (*Coen-Sanchez et al., 2022*). In this exploratory analysis we seek to understand the "research gap" between reproductive health and disease and other areas of medical research, and to suggest ways of closing this gap.

## Results

### Comparing numbers of publications

To benchmark research on reproductive health and disease, we used the PubMed database to compare the number of articles published on seven reproductive organs and seven non-reproductive organs between 1966 and 2021 (*Table 1*). While the reproductive organs are not essential to postnatal life, we posit that the placenta and the uterus are as essential to fetal survival in utero as the lungs and the heart are to postnatal survival after birth. Our analysis revealed that the average number of articles on non-reproductive organs was 4.5 times higher than the number on reproductive organs (and ranged between about 2 and 20 in pairwise comparisons). The reproductive organs with the most publications were the breast and prostate.

The research landscape can change over time and efforts to reduce gender bias in research might have had an impact on the volume of

**Table 1.** Total number of matching articles from PubMed for seven non-reproductive keywords and seven reproductive keywords for the period 1966–2021.

| Keyword | Total matching articles |
|---|---|
| Non-reproductive keywords | |
| Brain | 1,058,995 |
| Heart | 851,955 |
| Liver | 834,006 |
| Lung | 652,797 |
| Kidney | 451,177 |
| Intestine | 120,034 |
| Pancreas | 99,772 |
| Reproductive keywords | |
| Breast | 464,629 |
| Prostate | 197,736 |
| Ovary | 83,971 |
| Placenta | 57,076 |
| Uterus | 55,971 |
| Testes | 32,344 |
| Penis | 15,019 |

reproductive research, so we plotted the number of publications on the 14 organs as a function year between 1966 and 2021 (*Figure 1A*). Breast and prostate were the only reproductive organs to increase in publication at a rate similar to the kidney; the second least studied non-reproductive organ in our list. The intestine was the only non-reproductive organ to show similar publication rates to the other five reproductive organs. To investigate further, we compared disease-driven research versus research not related to disease.

### Comparing research related to disease and research not related to disease

In the 1970s, the National Institutes of Health (NIH) initiated a war on cancer, and the breast and prostate are both associated with sex-specific cancers. We reassessed publication data with the added search parameter "NOT cancer" to eliminate cancer-based research (*Figure 1B*). We observed a reduction of approximately 20% for most non-reproductive organs; however, the reduction for publication on the breast and prostate was about 80%, suggesting that most research on these organs is driven by an interest in cancer research rather than reproductive health and disease (*Figure 1B*).

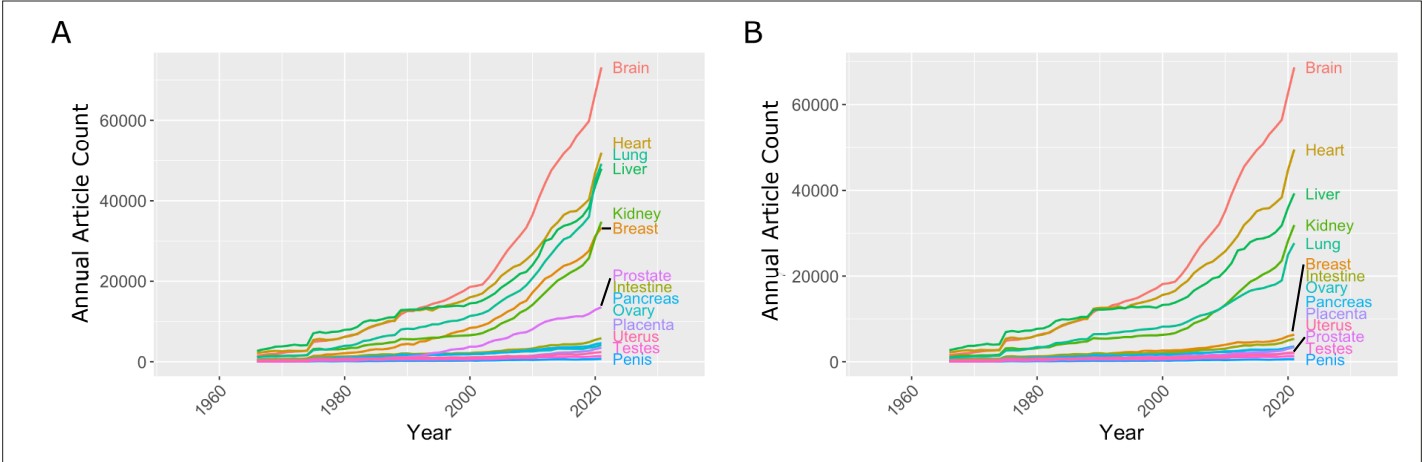

**Figure 1.** Number of articles published every year on seven reproductive organs and seven non-reproductive organs. (**A**) The number of articles published on most of the non-reproductive organs (including the brain, heart, lung and liver) has increased more rapidly than the number of articles published on the reproductive organs. (**B**) Removing articles that contain the keyword cancer has relatively little effect on the number of articles for non-reproductive organs (with the exception of the lung), but has a significant impact on the number of articles for the two reproductive organs with the most articles: the breast and prostate. Data extracted from PubMed using organ-specific keyword searches for the period 1966–2021.

The online version of this article includes the following source data for figure 1:

**Source data 1.** Articles per year for reproductive and non-reproductive organs, with and without the keyword cancer.

Then, for each organ, we plotted the number of publications related to disease on the vertical axis, and the number not related to disease on the horizontal axis, which revealed a high degree of variation among the organs (*Figure 2*). For three non-reproductive organs (brain, heart, and liver) the number of publications not related to disease was almost three times as high as the number related to disease, and for two non-reproductive organs (kidney and lung) the numbers were similar. For the breast and prostate, on the other hand, the number of publications related to disease was three times as high as the number not related to disease. For the five remaining reproductive organs, and also for the intestine and pancreas, the number of publications not related to disease was about twice as high as the number related to disease (although the total number of publications for these seven organs was about an order of magnitude lower than the number for the other seven organs).

### Research funding
Next we used databases belonging to the Canadian Institutes of Health Research (CIHR) and the NIH to investigate funding trends for the different organs. The 14 keywords (brain, heart, liver, lung, kidney, intestine, pancreas, breast, prostate, ovary, uterus, penis, testes, and placenta) were entered into each database, and we extracted funding data for the period between 2013 and 2018. These organs were

chosen as keywords to investigate the funding related to a basic understanding of the biology of these organs. Although grants that relate to pregnancy or fertility may not be captured, these topics are much broader and would introduce subtopics outside of the reproductive scope, similar to using keywords such as metabolism or behaviour. *Table 2* gives the number of projects for each keyword and the corresponding average funding amount per grant for the CIHR, and the same for the NIH. Our analysis found that the mean grant amounts for the CIHR and NIH are similar between different keyword research topics (CIHR: $ 370 000 ± $ 50 000; NIH: $ 481 500 ± $ 50 000). The similar funding amounts between different organs are encouraging and may result from standard funding guidelines for biomedical research. However, our analysis found that the average number of funded projects is much higher for non-reproductive organs compared to reproductive organs for both the CIHR (800 vs 115) and the NIH (31 000 vs 5 300).

## Discussion
Our analysis suggests a bias against research into reproductive health and disease, and it is important that efforts are made to eliminate this bias so that research into reproductive medicine does not fall further behind. The higher levels of research observed for some reproductive organs (notably the breast and prostate) were

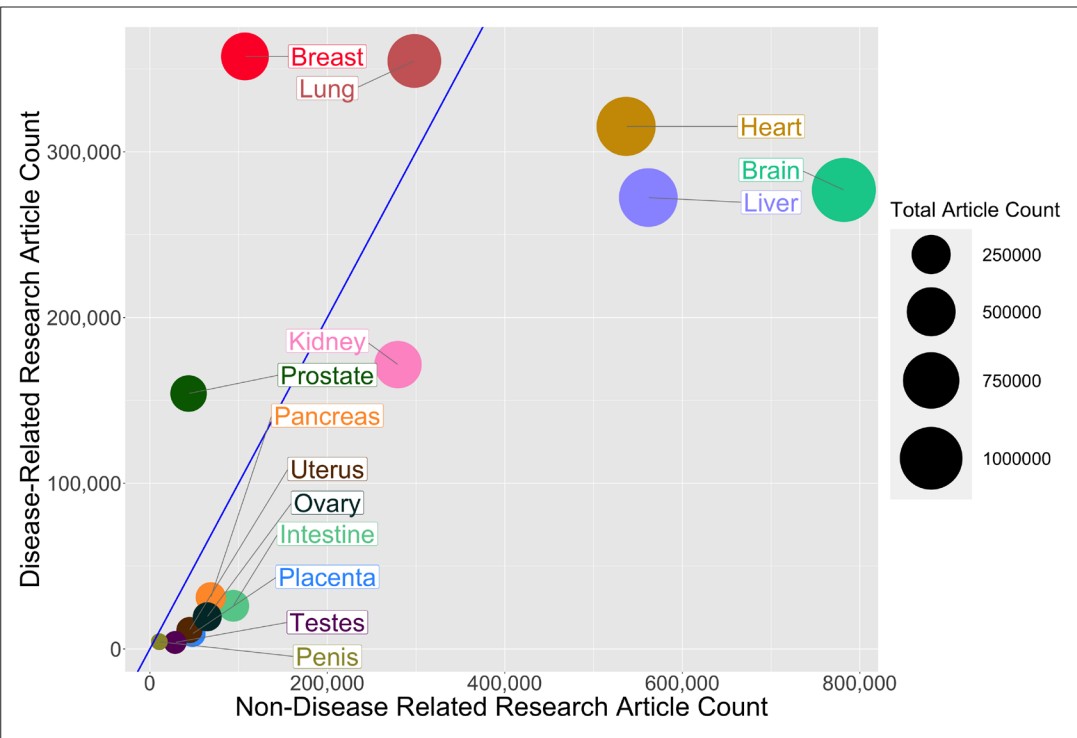

**Figure 2.** Comparing research related to disease and research not related to disease for reproductive and non-reproductive organs. For each organ (colored circles) the vertical axis shows the number of publications for the period 1966–2021 related to disease, and the horizontal axis shows the number not related to disease: the area of the circle is proportional to the total number of publications. The straight blue line corresponds to equal numbers of disease-related and non-disease-related publications, so organs to the right of this line (notably non-reproductive organs such as the brain, heart and liver) tend to be the subject of more basic or non-disease-related research, whereas organs to the left of this line (notably reproductive organs such as the breast and prostate) tend to be the subject of disease-related research. The lung is the only non-reproductive organ in our sample to the left of the blue line.

The online version of this article includes the following source data for figure 2:

**Source data 1.** Total number of articles on research related to disease and research not related to disease for reproductive and non-reproductive organs.

driven by cancer-focused research, but this has not led to an increase in the level of non-disease-related research on these organs (*Figure 1B*). Factors such as Breast Cancer Awareness Month (*Jacobsen and Jacobsen, 2011*) and screening programmes for prostate cancer (*Dickinson et al., 2016*) likely led to the increase in publications about these two reproductive organs.

While our analysis is suggestive that many reproductive organs achieve a good balance of non-disease versus disease-related research, the paucity of research is highly problematic to the field. An important consideration is that a lack of non-disease-related research on reproductive organs may hinder progress in diagnosing and treating a wide range of pathologies (including preeclampsia, polycystic ovary syndrome, and endometriosis).

In a competitive funding system, publications are correlated to successful grants and dollar values awarded. Across research areas, we found that the mean grant dollar amounts per project are similar. However, the numbers of funded research projects on non-reproductive organs were higher than the numbers for reproductive organs by a factor of 6–7 (which is slightly larger than the discrepancy seen in publication rates). An important consideration is that the part of the NIH that supports reproductive research in the US, the National Institute of Child Health and Development, is one of the lowest-funded institutes at the NIH and does not have the word reproduction in its title. In Canada, the Human Development, Child and Youth Health Institute of CIHR is a funder of most pregnancy and reproductive biology grants, typically awarded through

**Table 2.** Total number of projects funded and average grant (in Canadian or US dollars) for the Canadian Institutes of Health Research (columns 2 and 3) and the US National Institutes of Health (columns 4 and 5) for the years 2013–2018 for seven non-reproductive keywords and seven reproductive keywords (column 1).
Source data for this table is available in *Table 2—source data 1*.

| Keyword | Number of projects (CIHR) | Average grant funded (CAD) | Number of projects (NIH) | Average grant funded (USD) |
|---|---|---|---|---|
| Non-reproductive keywords | | | | |
| Brain | 1686 | $391,023 | 81666 | $441,149 |
| Heart | 1214 | $369,665 | 43833 | $491,993 |
| Liver | 1597 | $314,473 | 22072 | $454,276 |
| Lung | 526 | $371,154 | 34492 | $525,631 |
| Kidney | 347 | $424,360 | 21176 | $508,853 |
| Intestine | 128 | $444,490 | 5800 | $371,727 |
| Pancreas | 96 | $491,274 | 8649 | $482,901 |
| Reproductive keywords | | | | |
| Breast | 459 | $336,734 | 19132 | $525,134 |
| Prostate | 143 | $299,034 | 8960 | $514,638 |
| Ovary | 42 | $379,349 | 4814 | $520,804 |
| Placenta | 105 | $369,825 | 2169 | $526,147 |
| Uterus | 45 | $324,690 | 1356 | $509,250 |
| Testes | 10 | $372,110 | 340 | $500,160 |
| Penis | 1 | $304,676 | 323 | $369,434 |

The online version of this article includes the following source data for table 2:

**Source data 1.** Source data for *Table 2*.

the Clinical Investigation – A panel, and it may be that the use of a clinical panel to fund this area of research inhibits non-diseased focused research. This panel is well-funded relative to other panels; however, some research areas (e.g., cardiovascular and neurological research) have more than one panel.

A growing political and societal emphasis is placed on disease-related research, such as cancer. This may arise from a view of basic research as ineffective or inefficient compared to applied research (*Lee, 2019*). Perhaps this is best seen in our analysis by the high percentage of research publications on the prostate and breast that are due to cancer research, whereas most research on the other reproductive organs we studied was not disease-related. While the placenta and uterus are widely viewed as causal organs for reproductive complications that claim large numbers of maternal and neonatal lives, and treatments cost tens of billions of US dollars every year, there is relatively little disease-related

research into these organs. The investigation of cancer biology within a reproductive organ can rely on knowledge of cancer in other organ systems. However, the low levels of research into reproductive organs relative to other organs means that there is much less foundational knowledge to rely on when seeking to develop treatments for diseases of these organs. Moreover, there are fewer researchers who are experienced on working with these organs.

There are several limitations to our approach. One important limitation is that the number of unfunded grant applications is not accessible, so we could not determine if the lower numbers of grants for research on reproductive health and disease were due to proportionally lower total application numbers, or to a bias against reproductive research. Funding bodies should conduct internal analyses to determine appropriate action. The use of keywords to distinguish between non-disease and disease-related research is a limitation, and the relatively low numbers of

publications on reproductive organs can also present challenges when making comparisons. However, the differences we observe between research into reproductive and non-reproductive organs (as measured by numbers of publications and levels of funding) are large and are unlikely to result from missing search terms.

## Conclusions

How can we address the research gap and enable the field of reproductive health and disease to catch up with other areas of research? Based on our analysis, we need to increase the number of researchers working on reproductive organs and related pathologies. Recent efforts by the NIH, such as the Human Placenta Project (*Guttmacher et al., 2014*), indicate a recognition of the need to increase research capacity in reproductive sciences, and may lead to further increases in both interest and research capacity in the longer term.

New researchers may avoid the reproductive field due to social and political factors and the research gap (ie, the low levels of grant funding and publications), and this in turn may discourage students and trainees, which will make it even more difficult to increase the size of the research base. While continued advocacy, education, and political lobbying may help to overcome many of the social and political factors, closing the research gap will require other approaches.

To increase researchers and research output, we may learn lessons from the examples of breast and prostate cancer. In both cases, research increased dramatically from a historically low level. While public campaigns played a prominent role in these increases, the existence of a large pool of researchers and trainees already working on other types of cancers was probably more important (as it was these researchers, rather than those doing non-disease-related research on these organs, who did most of the work on breast and prostate cancer). However, this is unlikely to work for preeclampsia and other reproductive pathologies as there are no large pools of existing researchers available to switch the focus of their work.

Therefore, to increase research capacity, we should promote collaborations between researchers working on reproductive health and disease and those working in other areas of physiology and medicine, especially other areas with much higher research capacities. There are plenty of examples that show the benefit of such an integrated approach. For instance, female sex hormones protect against many aging diseases, such as cardiovascular and neurological diseases, leading to the prescription of hormone replacement therapies after menopause in some women (*Paciuc, 2020*).

Links to immunology, cardiology and other systems can be used to increase research capacity. During pregnancy, there are dramatic changes in maternal physiology, including metabolism, the immune system, and cardio-pulmonary systems, and consequently, these are the same systems affected by reproductive pathologies. Preeclampsia predisposes the mother to a long-term cardiovascular risk of developing peripheral artery disease, coronary artery disease, and congestive heart failure (*Rana et al., 2019*). Additionally, complications of the liver and kidney are associated with preeclampsia. Polycystic ovary syndrome and endometriosis are related to metabolism problems and the risk of cancer development. Children born from pregnancies affected by preeclampsia or fetal growth restriction are at a 2.5 times higher risk of developing hypertension and require anti-hypertensive medications as adults (*Ferreira et al., 2009*; *Fox et al., 2019*).

The pathological interaction of reproductive with non-reproductive systems and organs should attract investigators from nephrology, hepatology and cardiovascular research, where the total number of researchers is 10–20 times as high as the number in reproductive health and disease. If just 1% of the researchers in the cardiovascular field were to refocus on pregnancy-related cardiovascular adaptation and pathologies, this would increase reproductive research by 10%.

Our neglect of the placenta and reproductive biology impedes other biomedical research areas. In cancer research, the methylation patterns of tumours look most like those found in the placenta, but why placenta methylation patterns are so unlike all other organs is not known (*Smith et al., 2017*; *Rousseaux et al., 2013*). In regenerative medicine, the immune-modulating genes used by the placenta (*Szekeres-Bartho, 2002*) are repurposed to generate universally transplantable stem cells and tissues (*Han et al., 2019*). A poor understanding of reproductive biology is dangerous, considering emerging diseases that affect pregnancy and fetal development, such as the recent Zika virus outbreak (*Schuler-Faccini et al., 2016*; *Calvet et al., 2016*). There are likely many other broad benefits to better understanding reproductive biology. The time to act is now, as waiting longer will not improve the situation.

## Methods

### Publication rates

Published research manuscripts were searched in NCBI's PubMed database (https://pubmed.ncbi.nlm.nih.gov/) between and including the years 1966 and 2021. Keywords for each search pertained to a specific organ or disease and were limited to the title/abstract of the manuscripts. The organs used for these analyses were the brain, heart, liver, lung, kidney, intestine, pancreas, breast, prostate, ovary, uterus, penis, testes, and placenta. We restricted the organ publication timelines to the years 1966–2021 and extracted the annual article count. The organ publication timeline was reconducted with the addition of the search parameter "NOT cancer".

### Funding rates

Grant funding data was obtained from the CIHR funding database (https://webapps.cihr-irsc.gc.ca/funding/Search?p_language=E&p_version=CIHR) and the NIH reporter tool (https://reporter.nih.gov) by searching keywords in the title and abstracts/summary. Keywords used for these searches were brain, heart, liver, lung, kidney, intestine, pancreas, breast, prostate, ovary, uterus, penis, testes, and placenta. The years were restricted to 2013–2018. The total number of projects pertaining to each search term during this period was extracted, and the total amount of funding for those projects was averaged.

### Graphing

All graphs were produced using R (version 4.0.2) in R Studio (version 1.3.1073). R packages used were ggplot2, tidyverse, formattable, gridExtra, RColorBrewer, ggrepel.

### Acknowledgements

We thank the University of Toronto and the Department of Physiology for providing the opportunity and supporting the completion of this review. We also thank the librarians who offered expert advice on keyword searches of databases.

**Natalie D Mercuri** is in the Department of Physiology, University of Toronto, Toronto, Canada

ⓘ http://orcid.org/0000-0002-2276-5835

**Brian J Cox** is in the Department of Physiology and the Department of Obstetrics and Gynaecology, University of Toronto, Toronto, Canada

b.cox@utoronto.ca

ⓘ http://orcid.org/0000-0001-7146-6041

**Author contributions:** Natalie D Mercuri, Formal analysis, Funding acquisition, Visualization, Methodology, Writing – original draft; Brian J Cox, Conceptualization, Supervision, Writing – review and editing

**Competing interests:** The authors declare that no competing interests exist.

### Funding

| Funder | Grant reference number | Author |
|---|---|---|
| University of Toronto | | Natalie D Mercuri |
| Canada Research Chairs | | Brian J Cox |

The funders had no role in study design, data collection and interpretation, or the decision to submit the work for publication.

### Decision letter and Author response

Decision letter https://doi.org/10.7554/eLife.75061.sa1
Author response https://doi.org/10.7554/eLife.75061.sa2

## Additional files

### Supplementary files

• MDAR checklist

### Data availability

All data were obtained from public databases (PubMed/NCBI, NIH and CIHR). Source data files for Figure 1, Figure 2 and Table 2 are available (see figure and table captions for details).

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
