## [Decision Letter]

**Decision letter after peer review:**

Thank you for submitting the paper "A Poor Research Landscape Hinders the Progression of Knowledge and Treatment of Reproductive Diseases" for consideration by *eLife*. Your article has been reviewed by 3 peer reviewers, and the evaluation has been overseen by a Reviewing Editor and a Senior Editor. The following individuals involved in review of your submission have agreed to reveal their identity: Marleen van Gelder (Reviewer #1); James Roberts (Reviewer #3).

This article will need considerable revision to be suitable for publication as a Feature Article. In particular, you will need to address the concerns raised by the referees (see below), and also address a number of editorial points.

*Reviewer #1*

In this manuscript, Mercuri and Cox aimed to quantify the advancement of research in reproductive sciences relative to other medical disciplines. They compared two indicators of the research landscape: published research manuscripts and funded projects. The results showed lower publication rates for research on reproductive organs compared to selected non-reproductive organs, in particular concerning basic research. In addition, a relatively small number of grants was funded for projects on diseases with a reproductive focus. Based on these data, the authors concluded that the gap in knowledge and treatment of diseases of the reproductive organs is at least partially caused by a poor research landscape.

Although the conclusions of this paper are somewhat supported by the data, some aspects of the methods and reporting need to be clarified.

[Note: The following point is covered by the queries in the Word version I have sent you]

1) The manuscript, and in particular the Introduction and Discussion sections, could benefit from restructuring, in which adhering to a relevant reporting guideline may be helpful. For example, the authors provide relatively extensive background information on a number of important reproductive health disorders, but the level of detail does not contribute to setting the aim for the study. Moreover, the last paragraph of the introduction section (lines 92-100) already seems to include the conclusion of this paper.

[Note: Please address points b, d and f below. The other points are covered by the queries in the Word version I have sent you]

2) Concerns regarding the methods:

a) Citations in PubMed are known to be selective before 1966; consider using a fixed start date/year for the search.

b) The results strongly depend on the organs and diseases selected to be included in the 'reference group'. Provide a rationale for the selection of organs, which in the current analysis only seem to include major organs that are known to be well-studied, and not organs such as skin, eyes, intestine, pancreas, spleen or urinary bladder. The selection is vital for drawing robust conclusions from the data.

c) The approach to distinguish between basic and applied research is not validated.

d) The prevalence of diseases reported in Figure 4 is highly country-specific, in particular for tuberculosis. Therefore, this comparison may not be suitable for an international audience.

e) The most important limitation of the grant funding data was already mentioned: "the number and keywords of failed grant applications were not accessible" (lines 271-272). Therefore, it is hard to draw conclusions on failure of grant applications on reproductive health.

f) The rationale for the keywords used in the funding databases is missing and likely to yield selective results. Many reproductive health related projects may be missed, as keywords such as pregnancy and subfertility were not included. And also in this search, the selection of keywords for the reference group seems biased.

[Note: Please consider adding a table as suggested below; however, this is optional rather than essential.]

3) To emphasize the lack of knowledge in relation to disease burden, a table summarizing the prevalence, number of publications, and grants could summarize the results.

[Note: This point is covered by the queries in the Word version I have sent you]

4) A number of topics and statements in the Discussion section seem to be unrelated to the aim of this study. Examples include the female representation in STEM disciplines and the correlation between research publications and changes in policy (this was not specifically analyzed and would require additional analyses).

*Reviewer #2*

[Note: Please address the following point]

While the authors have attempted to be broad in their assessment of reproduction research, they seem to neglect two very broad areas of women's health for which there is little research: menstruation and menopause. Both are only mentioned in the discussion, and referenced with respect to promotion of the study of human physiology. Given the focus on lack of basic understanding of reproductive organs, it may be worth mentioning these, particularly in comparison to the depth of research on erectile dysfunction; this may also help to emphasize the fact that the lack of research in reproduction primarily affects women (though there are of course consequences for men's health, including the period in the womb).

[Note: This point is covered by the queries in the Word version I have sent you]

Figure 1: the color code is not clear; Not sure how this could be better represented, but maybe listing the organs from high to low for both parts a and b in the legend? Or magnifying one part of each graph? In particular, the 80% loss of publications in breast/prostate when applying the search term "NOT cancer" does not come through; so perhaps a graph focusing on just these two organs showing the original search and the "NOT cancer" search results would be best?

[Note: This point is covered by the queries in the Word version I have sent you]

Tables 2 and 3: It is not clear how this search was done; was the project title or abstract of grants searched for these key terms?

[Note: Please address the following point]

Discussion (including lines 259-260): I'm not sure that the conclusion drawn here is consistent with the data? The authors somewhat confusingly alternate between lack of research in reproduction as a whole vs. lack of basic research in this area.

[Note: Addressing the following point is optional, not essential.]

Another point of discussion that merits mention here is how the lack of interest/emphasis on reproduction research by funding agencies in turn affects the perception of "impact" of such research: i.e. both in terms of how low impact factors of reproduction journals are compared to journals in other fields, but also how the high-impact journals (Cell/Science/Nature) view/receive submissions from researchers in this area.*Reviewer #3*

The authors propose that research in reproductive areas lags behind that of other areas of biology. They support this with information from publications and funding sources.

This is a presentation of importance to investigators in all fields, funders and the general public. For reproductive investigators it provides objective data to support the lagging of reproductive research and to investigators in other areas of biology and the general public should be an eye opening demonstration of the huge gap between research in reproduction and other areas of biology. One would hope it would also provide a motivation to funders to modify the situation.

The authors remind us of the importance of reproduction on the survival of the species and provide extensive data on specific examples of the impact of reproductive diseases. They then use review of publications keyed to reproductive organs and non-reproductive organs both currently and over time. They point out that research on non-reproductive organs is 5 to 20 times more frequent than that on reproductive organs. [Note: Please address the point made in the following sentence] They should make it clearer that this is referring to specific organs and not a comparison sum of research on all organs of reproduction and not reproduction. They show that over time this discrepancy has increased with the exception of prostate, and breast research but even with those it is evident this is research related specifically to cancer and not normal organ function.

They make a slightly less compelling comparison on the portion of research devoted to basic understanding or clinical research which for nonreproductive organs is considerably more for basic science than in reproductive organs. [Note: Please address the point made in the following sentence] However, this is likely compromised by the relative minute number of either type of studies in reproduction.

They then make comparisons between the impact of specific reproductive topics and publications. They state that although preeclampsia and breast cancer have a similar prevalence the number of breast cancer publications are much higher. [Note: Please address the point made in the following sentence] To me the comparison of a disorder with high mortality (breast cancer) and far lower mortality (preeclampsia) does not provide a compelling argument and also is a little off target for comparing reproductive and nonreproductive research.

[Note: Please address the point made in the following paragraph]

They make a similar comparison of PCOS a reproductive disorder with other non-reproductive disorders of similar or lower prevalence, autism, tuberculosis, Crohn's Disease and Lupus with a much lower publication rate for PCOS. Again, this seems a bit of comparing apples and oranges.

They investigate the relative funding of research on these topics in the US and Canada and find that the size of individual grants for reproductive and non-reproductive research in both countries is similar but that the number of funded grants for specific non-reproductive organs is, that like that of publications, is about 2 to 20 times higher for nonreproductive organs.

The authors present their conclusions of the reason for the discrepancy. They point out gender bias which has been a target for improvement for several years and has been reduced but research is still not on an equal basis for men and women. However, the bias goes beyond gender since male reproductive research publications and funding also lags. They conclude that there is a general bias against reproductive research. [Note: Please consider mentioning the following point in your article] Interestingly they do not cite a major support for this conclusion, that the major NIH institute supporting reproductive research, the National Institute of Child Health and Development (NICHD)is one lowest funded institutes and does not have reproduction in its title.

They provide two general suggestions to increase reproductive research. The first is to increase funding and the second to involve other forms of research in studies supporting the role of reproductive disorders and physiology in non-reproductive studies. [Note: Please address the point made in the rest of this paragraph] They point out the relationship of preeclampsia to later life cardiovascular disease as an example of this. Unfortunately, they state this relationship as causal which has not been established. Nonetheless studying preeclampsia will likely provide information useful to cardiovascular health.

It is possible that linking publications and funding amounts to conclusions about bias against reproductive research is not precise. However, the magnitude of the differences strongly supports the authors' premise.

This interesting presentation makes and important point about the fact that reproductive research lags beyond other biological research. They do this through the use of publication and grant funding reviews. The differences are large in a direction that support the point they are making. There are some suggestions that I believe would improve the presentation.

[Note: Please address the following three points]

1. There should be a bit more discussion of the limitations of their approach.

2. In the comparisons of disorders of reproduction and non-reproduction they should indicate the limitations of comparing very different disorders.

3. Preeclampsia as a cause of later life CVD has not been established. They are related.

---

## [Author Response]

Reviewer #1In this manuscript, Mercuri and Cox aimed to quantify the advancement of research in reproductive sciences relative to other medical disciplines. They compared two indicators of the research landscape: published research manuscripts and funded projects. The results showed lower publication rates for research on reproductive organs compared to selected non-reproductive organs, in particular concerning basic research. In addition, a relatively small number of grants was funded for projects on diseases with a reproductive focus. Based on these data, the authors concluded that the gap in knowledge and treatment of diseases of the reproductive organs is at least partially caused by a poor research landscape.Although the conclusions of this paper are somewhat supported by the data, some aspects of the methods and reporting need to be clarified.[Note: The following point is covered by the queries in the Word version I have sent you]1) The manuscript, and in particular the Introduction and Discussion sections, could benefit from restructuring, in which adhering to a relevant reporting guideline may be helpful. For example, the authors provide relatively extensive background information on a number of important reproductive health disorders, but the level of detail does not contribute to setting the aim for the study. Moreover, the last paragraph of the introduction section (lines 92-100) already seems to include the conclusion of this paper.

This query has been responded to the in Word file

[Note: Please address points b, d and f below. The other points are covered by the queries in the Word version I have sent you]2) Concerns regarding the methods:a) Citations in PubMed are known to be selective before 1966; consider using a fixed start date/year for the search.

This query has been responded to the in Word file. We have now used a fixed date of 1966 as the early timepoint and as indicated in the Word file.

b) The results strongly depend on the organs and diseases selected to be included in the 'reference group'. Provide a rationale for the selection of organs, which in the current analysis only seem to include major organs that are known to be well-studied, and not organs such as skin, eyes, intestine, pancreas, spleen or urinary bladder. The selection is vital for drawing robust conclusions from the data.

Organs such as brain, heart and lungs are essential for life. The placenta is similarly essential. Other organs such as kidney and liver are also essential but not as immediate. We now include the intestine as a reference point.

Our preliminary analysis found that Skin has over 800,000 publication mentions, but it is not clear if this is the skin organ or a skin on something more work to eliminate background skin hits would be needed. Epidermis has 60,000 hits that are likely more specific, but we did find may abstracts and titles on the skin organ that do not use epidermis. Eyes are nearly 700,000 publications, intestine also over 700,000, pancreas has over 200,000 spleen is also over 200,000 urinary bladder has 130,000, which is similar to the placenta at just over 100,000

This preliminary search seems to still support our conclusion that placenta and reproductive organs are under-researched and only add a list of other organs that are better studied.

c) The approach to distinguish between basic and applied research is not validated.

This query has been responded to the in Word file

d) The prevalence of diseases reported in Figure 4 is highly country-specific, in particular for tuberculosis. Therefore, this comparison may not be suitable for an international audience.

Comparisons of diseases has been removed from the manuscript.

e) The most important limitation of the grant funding data was already mentioned: "the number and keywords of failed grant applications were not accessible" (lines 271-272). Therefore, it is hard to draw conclusions on failure of grant applications on reproductive health.

This query has been responded to the in Word file

f) The rationale for the keywords used in the funding databases is missing and likely to yield selective results. Many reproductive health related projects may be missed, as keywords such as pregnancy and subfertility were not included. And also in this search, the selection of keywords for the reference group seems biased.

We have removed disease focused terms form the search to ensure we capture organ focus research. The inclusion of pregnancy or subfertility would be misleading as it would include disciplines such as sociology and psychology. This is akin to searching for diabetes or metabolism to understand the research landscape on the pancreas.

3) To emphasize the lack of knowledge in relation to disease burden, a table summarizing the prevalence, number of publications, and grants could summarize the results.

We felt the separate tables made the information more digestible.

4) A number of topics and statements in the Discussion section seem to be unrelated to the aim of this study. Examples include the female representation in STEM disciplines and the correlation between research publications and changes in policy (this was not specifically analyzed and would require additional analyses).

This query has been responded to the in Word file. We have extensively edited and redrafted the Discussion section.

Reviewer #2[Note: Please address the following point]While the authors have attempted to be broad in their assessment of reproduction research, they seem to neglect two very broad areas of women's health for which there is little research: menstruation and menopause. Both are only mentioned in the discussion, and referenced with respect to promotion of the study of human physiology. Given the focus on lack of basic understanding of reproductive organs, it may be worth mentioning these, particularly in comparison to the depth of research on erectile dysfunction; this may also help to emphasize the fact that the lack of research in reproduction primarily affects women (though there are of course consequences for men's health, including the period in the womb).Figure 1: the color code is not clear; Not sure how this could be better represented, but maybe listing the organs from high to low for both parts a and b in the legend? Or magnifying one part of each graph? In particular, the 80% loss of publications in breast/prostate when applying the search term "NOT cancer" does not come through; so perhaps a graph focusing on just these two organs showing the original search and the "NOT cancer" search results would be best?

These corrections have been made to the in Word file.

Tables 2 and 3: It is not clear how this search was done; was the project title or abstract of grants searched for these key terms?

These corrections have been made to the in Word file.

Discussion (including lines 259-260): I'm not sure that the conclusion drawn here is consistent with the data? The authors somewhat confusingly alternate between lack of research in reproduction as a whole vs. lack of basic research in this area.

We agree and have focused the discussion on the general low level of publications and low level of researchers in the field.

Another point of discussion that merits mention here is how the lack of interest/emphasis on reproduction research by funding agencies in turn affects the perception of "impact" of such research: i.e. both in terms of how low impact factors of reproduction journals are compared to journals in other fields, but also how the high-impact journals (Cell/Science/Nature) view/receive submissions from researchers in this area.

This is an issue many discipline struggle with. A low number of researchers in a field tends to create low levels of impact as measured through citations. Attempts to normalize impact factors and citation rates to the size of the field may help. While we agree with the reviewers comments we cannot address within our study.

Reviewer #3The authors propose that research in reproductive areas lags behind that of other areas of biology. They support this with information from publications and funding sources.This is a presentation of importance to investigators in all fields, funders and the general public. For reproductive investigators it provides objective data to support the lagging of reproductive research and to investigators in other areas of biology and the general public should be an eye opening demonstration of the huge gap between research in reproduction and other areas of biology. One would hope it would also provide a motivation to funders to modify the situation.The authors remind us of the importance of reproduction on the survival of the species and provide extensive data on specific examples of the impact of reproductive diseases. They then use review of publications keyed to reproductive organs and non-reproductive organs both currently and over time. They point out that research on non-reproductive organs is 5 to 20 times more frequent than that on reproductive organs. [Note: Please address the point made in the following sentence] They should make it clearer that this is referring to specific organs and not a comparison sum of research on all organs of reproduction and not reproduction. They show that over time this discrepancy has increased with the exception of prostate, and breast research but even with those it is evident this is research related specifically to cancer and not normal organ function.

Thank you for this comment. These clarifications have been made to the in Word file.

They make a slightly less compelling comparison on the portion of research devoted to basic understanding or clinical research which for nonreproductive organs is considerably more for basic science than in reproductive organs. [Note: Please address the point made in the following sentence] However, this is likely compromised by the relative minute number of either type of studies in reproduction.

We agree that the lower level make estimating the ratio of basic to applied very challenging. But there seems to be a tendency to bias to basic research. We made some changes to the results and discussion to acknowledge this challenge.

They then make comparisons between the impact of specific reproductive topics and publications. They state that although preeclampsia and breast cancer have a similar prevalence the number of breast cancer publications are much higher. [Note: Please address the point made in the following sentence] To me the comparison of a disorder with high mortality (breast cancer) and far lower mortality (preeclampsia) does not provide a compelling argument and also is a little off target for comparing reproductive and nonreproductive research.

We agree and have remove the section discussing a comparison of disease prevalence and mortalities. We realize there was no benefit to comparison disease prevalence and severity.

They make a similar comparison of PCOS a reproductive disorder with other non-reproductive disorders of similar or lower prevalence, autism, tuberculosis, Crohn's Disease and Lupus with a much lower publication rate for PCOS. Again, this seems a bit of comparing apples and oranges.

We agree and have remove the section discussing a comparison of disease prevalence and mortalities. We realize there was no benefit to comparison disease prevalence and severity.

They investigate the relative funding of research on these topics in the US and Canada and find that the size of individual grants for reproductive and non-reproductive research in both countries is similar but that the number of funded grants for specific non-reproductive organs is, that like that of publications, is about 2 to 20 times higher for nonreproductive organs.The authors present their conclusions of the reason for the discrepancy. They point out gender bias which has been a target for improvement for several years and has been reduced but research is still not on an equal basis for men and women. However, the bias goes beyond gender since male reproductive research publications and funding also lags. They conclude that there is a general bias against reproductive research. [Note: Please consider mentioning the following point in your article] Interestingly they do not cite a major support for this conclusion, that the major NIH institute supporting reproductive research, the National Institute of Child Health and Development (NICHD)is one lowest funded institutes and does not have reproduction in its title.

Thank you for this comment, we have added it!

They provide two general suggestions to increase reproductive research. The first is to increase funding and the second to involve other forms of research in studies supporting the role of reproductive disorders and physiology in non-reproductive studies. [Note: Please address the point made in the rest of this paragraph] They point out the relationship of preeclampsia to later life cardiovascular disease as an example of this. Unfortunately, they state this relationship as causal which has not been established. Nonetheless studying preeclampsia will likely provide information useful to cardiovascular health.

Thank you for the comment, we modified our statement to an observed increased risk of cardiovascular disease, as the risk may be causal or associated as the reviewer stated.

It is possible that linking publications and funding amounts to conclusions about bias against reproductive research is not precise. However, the magnitude of the differences strongly supports the authors' premise.This interesting presentation makes and important point about the fact that reproductive research lags beyond other biological research. They do this through the use of publication and grant funding reviews. The differences are large in a direction that support the point they are making. There are some suggestions that I believe would improve the presentation.1. There should be a bit more discussion of the limitations of their approach.

We have added more caveats about our approach and interpretation

2. In the comparisons of disorders of reproduction and non-reproduction they should indicate the limitations of comparing very different disorders.

The comparisons of diseases has been removed.

3. Preeclampsia as a cause of later life CVD has not been established. They are related.

This is addressed as per the above comment.